# Immunopeptidomics-based design of mRNA vaccine formulations against *Listeria monocytogenes*

Rupert L. Mayer [1,2,3,9], Rein Verbeke [4,5], Caroline Asselman [1,2,6], Ilke Aernout [4,5], Adillah Gul[1,2], Denzel Eggermont [1,2], Katie Boucher[1,2,3], Fabien Thery[1,2], Teresa M. Maia [1,2,3], Hans Demol[1,2,3], Ralf Gabriels[1,2], Lennart Martens[1,2], Christophe Bécavin[7], Stefaan C. De Smedt [4,5], Bart Vandekerckhove[5,8], Ine Lentacker [4,5] ✉ & Francis Impens [1,2,3] ✉

*Listeria monocytogenes* is a foodborne intracellular bacterial pathogen leading to human listeriosis. Despite a high mortality rate and increasing antibiotic resistance no clinically approved vaccine against *Listeria* is available. Attenuated *Listeria* strains offer protection and are tested as antitumor vaccine vectors, but would benefit from a better knowledge on immunodominant vector antigens. To identify novel antigens, we screen for *Listeria* peptides presented on the surface of infected human cell lines by mass spectrometry-based immunopeptidomics. In between more than 15,000 human self-peptides, we detect 68 *Listeria* immunopeptides from 42 different bacterial proteins, including several known antigens. Peptides presented on different cell lines are often derived from the same bacterial surface proteins, classifying these antigens as potential vaccine candidates. Encoding these highly presented antigens in lipid nanoparticle mRNA vaccine formulations results in specific CD8[+] T-cell responses and induces protection in vaccination challenge experiments in mice. Our results can serve as a starting point for the development of a clinical mRNA vaccine against *Listeria* and aid to improve attenuated *Listeria* vaccines and vectors, demonstrating the power of immunopeptidomics for next-generation bacterial vaccine development.

*Listeria monocytogenes* (further referred to as *Listeria*) is a major foodborne pathogen causing listeriosis in vulnerable individuals. Infection typically occurs by consumption of contaminated food such as unpasteurized cheese or meat products[1]. The bacterium's ability to grow at refrigerator temperatures renders it a considerable risk factor in food industry and demands high levels of hygiene and monitoring[2]. Particularly for immunocompromised individuals, elderly people and pregnant women *Listeria* poses a substantial threat[3]. In severe cases, the pathogen can lead to invasive gastroenteritis, sepsis, encephalitis, meningitis or endocarditis, while in pregnant women infection may lead to abortion and fetal loss[3].

After ingestion, *Listeria* can cross the intestinal barrier and via the lymph nodes enter the bloodstream from where it spreads to liver and spleen[4]. Once in the bloodstream, *Listeria* may also cross the blood–brain barrier[5] or the fetoplacental barrier[6] to inflict the aforementioned complications. As a facultative intracellular pathogen, *Listeria* is capable to hide from the humoral immune system by invading host cells. The bacterium can induce its uptake in host cells via receptor-mediated endocytosis involving its surface-expressed virulence factors internalin (Inl) A and InlB[7]. Once inside the endocytic vesicle, *Listeria* secretes phospholipase (Plc) A and PlcB as well as the pore-forming toxin listeriolysin O (LLO) to access the cytosol[8], where it

can replicate and spread to neighboring cells via actin-based motility and expression of the actin assembly-inducing protein (ActA) at the bacterial pole[9].

While human listeriosis is not a very common type of acute infection, it has a high fatality rate of up to 30% and case numbers are gradually increasing[1]. In 2017, 2502 cases of listeriosis were confirmed in the EU/EEA countries, while in 2007 only 1635 cases had been reported[10–12]. In Germany, the number of deceased listeriosis patients per year has risen considerably in recent years, and also more infections with antibiotic-resistant *Listeria* strains have been reported[13,14]. Often resulting from local outbreaks[15–17], it was estimated that in 2010 listeriosis caused 23,150 global sicknesses leading to 5463 deaths[18]. As *Listeria* occurs ubiquitously in the environment, domestic ruminants such as cattle, sheep and goats also get infected resulting in neurological and maternal–fetal listeriosis[19,20], hampering agricultural productivity and resulting in economic losses[21–23]. Even though cases of listeriosis are rising, no vaccines against *Listeria* are currently available or in clinical trials, presumably due to the still rather infrequent occurrence of symptomatic *Listeria* infections. Despite this restrained commercial interest to date, academic efforts to develop a safe and affordable vaccine are ongoing and could ensure the protection of vulnerable populations like pregnant women, elderly people or immunosuppressed patients[24–27]. Embracing an intracellular lifestyle, immune clearance of *Listeria* heavily relies on CD8+ T cell-mediated cytotoxicity and therefore an effective vaccine must be able to induce potent cellular immunity[28]. In comparison to clinically more problematic intracellular bacteria such as *Mycobacterium tuberculosis*, *Listeria* is relatively easy to cultivate and safe to work with and is therefore often used as a model system for intracellular bacterial infections[3].

So far mostly live attenuated vaccines against *Listeria* have been explored, typically resulting in high levels of protection in animal models[24–26,29,30]. Similarly, inactivated *Listeria* or bacterial ghosts (bacteria depleted of intracellular content) were explored as preclinical vaccine candidates[31,32]. Attenuated vaccines face a few challenges including genetic instability over extended periods of time rendering an attenuated strain more virulent again[33,34]. Next to attenuated strains, also cell-based, DNA-based, viral vector, subunit, and recombinant protein vaccines have been explored against listeriosis[35–39]. Antigens of *Listeria* that have proven to facilitate protective immunity include predominantly LLO and invasion-associated protein p60 (p60/iap), but also glyceraldehyde-3-phosphate dehydrogenase (GAPDH/gap)[38–40]. Utilizing only such a limited antigen pool is convenient as LLO and p60 are particularly well studied, but also presents the risk of protecting only part of the population since different MHC alleles (haplotypes) might favor presentation of different epitopes and antigens. Next to preventing listeriosis, attenuated *Listeria* strains are also tested as vaccine vectors expressing cancer-associated antigens[27], and numerous clinical studies are underway utilizing *Listeria* as vector to deliver tumor antigens for treatment of malignancies such as lung, prostate, brain, cervical cancer, and others[41,42]. A recent study however identified that immunodominant *Listeria* vector epitopes can strongly bind to host MHC molecules, thereby competitively inhibiting the presentation of the cargo cancer antigen and reducing the therapeutic effect[42]. Hence, knowledge about immunodominant *Listeria* epitopes could be critical to further ameliorate attenuated *Listeria* strains as cancer vaccine vectors.

The SARS-CoV-2 pandemic has clearly demonstrated the safety and effectiveness of messenger RNA (mRNA) vaccines and confirmed their role as next-generation vaccines[43,44]. In contrast to other vaccine platforms, these vaccines contain nucleoside-modified mRNA encoding pathogen antigens complexed within lipid nanoparticles (LNPs). The latter protects the mRNA and allows efficient uptake and translation of the encoded pathogen antigens by host cells to elicit both cellular and humoral immune responses[45]. Besides viral applications, mRNA-based vaccines hold great potential also for intracellular bacterial pathogens as safe and versatile platforms that might greatly accelerate the vaccine development and market rollout process. In contrast to viral pathogens, however, bacteria typically express several thousand proteins, which renders the task of choosing the right protein antigens for vaccination a daunting one. Despite the high potential of mRNA vaccines for (intracellular) bacteria, only a handful of studies have investigated this promising avenue to date[46–48].

Since cellular immunity and cytotoxic T cells are key to protect against intracellular pathogens, elucidating the antigens presented by MHC class I molecules on the surface of infected cells is critical for successful vaccine development against these pathogens. This can be achieved by mass spectrometry (MS)-based immunopeptidomics, a technology originally co-developed by Donald Hunt and Hans-Georg Rammensee[49–51]. While in the early days, technical limitations allowed the detection of only a handful of bacteria-derived peptides[52,53], mass spectrometry and analysis algorithms have evolved substantially now allowing to detect dozens of bacterial immunopeptides in a single analysis[54–57]. None of these recent immunopeptidomics studies however investigated the MHC class I immunopeptidome of *Listeria monocytogenes*, and the list of known *Listeria* antigens is rather limited. Only 206 peptides from 79 *Listeria* antigens are listed in the Immune Epitope Database (IEDB), mainly derived from LLO (69 peptides) and p60 (41 peptides) as well as from plcB, gap, mpl, prfA, and lmo0209 (three peptides each)[58]. Of these, 116 peptides are presented on MHC class I molecules.

To extend the antigen knowledge on *Listeria* we here applied an immunopeptidomics pipeline on two infected human epithelial cell lines. In between more than 15,000 human self-peptides, we identified 68 *Listeria* immunopeptides from 42 different bacterial antigens. Along with several known antigens, many novel antigens were detected, often derived from the bacterial periphery. Encoding highly presented antigens as vaccine candidates in mRNA vaccine formulations significantly reduced the bacterial load in liver and spleen in a vaccination-challenge study in mice. The results of this study can be used to improve *Listeria* vaccine vectors or for further preclinical development of an mRNA vaccine against *Listeria*, acting as a blueprint for the MS-based development of mRNA vaccines against other intracellular bacterial pathogens.

## Results

### MHC class I peptides presented on *Listeria*-infected cells

To identify novel *Listeria* antigens, we isolated MHC I presented peptides from cultured human HeLa and HCT-116 cells infected or not with *Listeria monocytogenes* EGD at a multiplicity of infection (MOI) of 50. Each condition was analyzed in four biological replicates starting from 350 to 540 million cells per replicate. Isolated immunopeptides from each replicate were split to subject one half to label-free liquid chromatography-tandem mass spectrometry (LC-MS/MS) analysis, while the other half was labeled with tandem mass tags (TMT), pooled, and pre-fractionated prior to LC-MS/MS analysis (Fig. 1a). TMT labeling comprises tagging of the peptides of each sample with different isobaric tags before LC-MS/MS analysis of the pooled sample, allowing relative quantification upon peptide fragmentation[59]. TMT labeling of immunopeptides has previously been shown to extend the comprehensiveness of immunopeptide identification by improving peptide ionizability and fragment ion intensity during LC-MS/MS analysis[60].

Following spectral identification with the PEAKS software and filtering for high confident hits with a false discovery rate (FDR) of 1%, in total we detected 15,767 host- and 84 *Listeria*-derived immunopeptides (Fig. 1b and Supplementary Data 1). Assessment of the peptide length distribution resulted in the expected predominance of 9mers among all identified peptides, and also a considerable proportion of 8mer, 10mer, 11mer, and 12mer peptides, in line with previous reports (Fig. 1c)[56,61,62]. Submission of 9mer sequences from both HeLa and

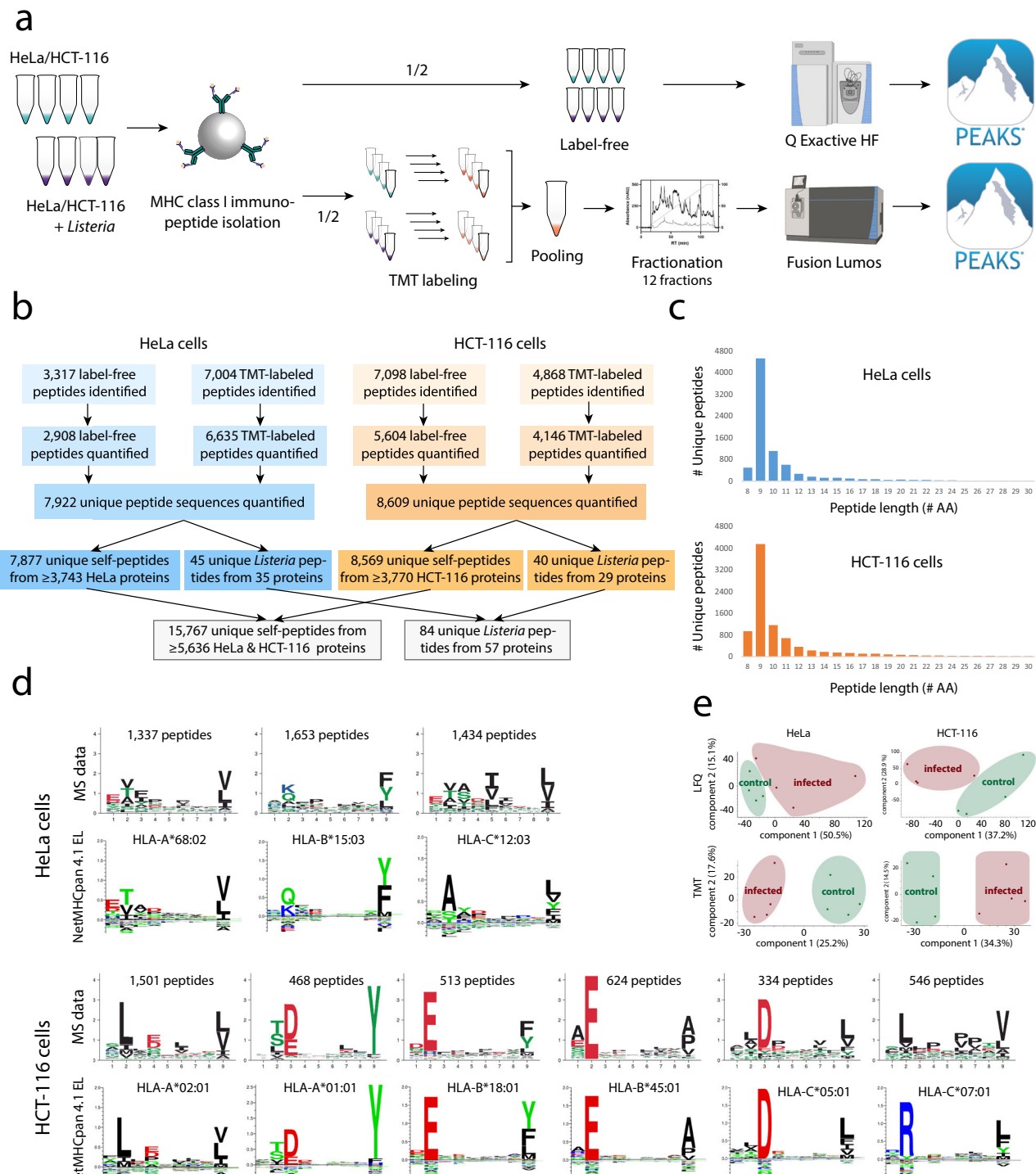

**Fig. 1 | Identification of MHC class I immunopeptides presented on *Listeria*-infected HeLa and HCT-116 cells. a** Four replicates of *Listeria*-infected and uninfected HeLa or HCT-116 cells were dissolved using a mild lysis buffer and purified as described[61]. One-half of the resulting immunopeptides were analyzed by label-free LC-MS/MS analysis on a Q Exactive HF mass spectrometer, while the other half was differentially labeled by TMT, pooled per cell line and fractionated into 12 fractions prior to LC-MS/MS analysis on a Fusion Lumos. Database searching was carried out using PEAKS Studio X+ ™ (version 10.5 build 20200219)[123,126]. Created with BioRender.com. **b** Overview of identified and quantified immunopeptides per cell line and in total (Supplementary Data 1). **c** Immunopeptide length distribution representing the typical dominance of 9mers in the human MHC I-presented immunopeptidome. **d** 9mer peptide sequences were submitted to MixMHCp 2.1 for motif deconvolution using sequence logos for visualization of the modeled position

weight matrices (PWMs) after unsupervised clustering[64,65]. To this end, the tool assumes that all peptides are of length equal to core length, are naturally aligned and that defined positions at the beginning (first three amino acids) and end (last two positions) can be found in the peptide sequence, which is typically the case for HLA-I ligands. HLA binding motifs derived from experimental data demonstrated good overall matching with the expected cell line-specific NetMHCpan 4.1 reference motifs[66,132,133]. For HeLa cells, two out of three motifs could be fully restored, while the motif for HLA-C*12:03 was only partially recovered. For HCT-116, five out of six HLA motifs were fully reconstituted, while HLA-C*07:01 was only incompletely recovered, likely due to typically low expression levels of many HLA-C alleles[61]. **e** Principal component analysis (PCA) using the immunopeptide intensities separated the uninfected samples from the *Listeria*-infected samples.

HCT-116 cells to MixMHCp 2.1 resulted in the reconstitution of most cell line-specific HLA motifs (Fig. 1d)[63–66]. For HeLa-cell derived immunopeptides, two out of the three expressed HLA alleles matched the reference motifs well[67]. For HCT-116 cells, five out of the six expressed HLA alleles were fully reconstituted with only HLA-C*07:01 remaining unresolved supposedly due to the typically lower expression levels of HLA-C alleles[61]. Principal component analysis of the immunopeptide intensities clearly separated the infected and uninfected replicate samples in both cell lines, indicating that *Listeria* infection has an important effect on the MHC Class I presented immunopeptidome (Fig. 1e). In contrast, *Listeria* infection did not lead to detectable upregulation of MHC Class II molecules on HeLa cells (Supplementary Fig. 1), together with negligible predicted binding affinities for *Listeria*-derived peptides (Supplementary Data 2) excluding significant contamination of MHC Class II-derived immunopeptides. Taken together, we identified more than 15,000 MHC class I peptides presented on *Listeria*-infected cells. The peptide length distribution and clustering into expected HLA binding motifs supported bona fide detection of the immunopeptides and high quality of the dataset.

### High confident detection of *Listeria* immunopeptides
Next, we compared immunopeptide abundancies between the infected samples and the uninfected controls. We used this quantitative comparison as an additional filtering step to select only high-confidence *Listeria* peptides. Indeed, bacterial immunopeptides are only expected in the infected samples, however, some of these peptides came with intensity values in the control samples suggesting they are likely false positive identifications (note that with an FDR of 1% we still expect about 160 false positive immunopeptides in the total dataset) or quantifications (i.e., matching of noise peaks). We therefore only selected *Listeria* peptides for further analysis that were (i) quantified by the PEAKS software in at least two of the infected samples and (ii) showed a higher average abundance in the infected samples compared to the uninfected control. Furthermore, *Listeria* peptide sequences were searched against the human database with all possible leucine/isoleucine permutations since leucine and isoleucine residues are virtually indistinguishable by mass spectrometry due to their identical mass. In addition, also sequences found in the human database were removed from the bacterial immunopeptide list. These filtering steps resulted in 68 high confident *Listeria* peptides originating from 42 bacterial proteins (Fig. 2a and Supplementary Data 2). 58 of these peptides were predicted by NetMHCpan EL 4.1 to bind to at least one of the HLA alleles expressed on HeLa or HCT-116 cells (Fig. 2b) with most peptides binding to HLA-A alleles (Supplementary Fig. 2), further supporting their high confident detection[67]. Moreover, we synthesized 24 of the *Listeria*-derived immunopeptides to compare their experimentally recorded MS2 spectrum with the spectrum of their synthetic counterpart. All synthetic and experimental spectra displayed a high degree of overlap with a Pearson correlation coefficient of >0.90, confirming correct bacterial immunopeptide identification (Fig. 2c and Supplementary Fig. 3).

As expected, *Listeria* peptides were amongst the most highly induced peptides presented in the infected samples (Fig. 2d) and absent in the control samples as observed from their missing (or occasional noise) values in the label-free data and low-intensity values in the TMT data (resulting from well-documented peptide co-isolation and ratio suppression) (Fig. 2e)[68,69]. Among the *Listeria* peptides identified in this project, only VAYGRQVYL from LLO was previously reported and listed in the IEDB. For two other LLO peptides (KIDYDDEMAY and SESQLIAKFGTA), prolonged sequences (AKIDYDDEMAYS, KIDYDDEMAYSESQ, KIDYDDEMAYSESQLIAKFGTAFK, DEMAYSESQLIAKFGTAFK, SESQLIAKFGTAFK) were identified in previous MHC class II studies[70,71]. Among the protein antigens of origin, eight are described in the IEDB, including many proteins from the prfA-virulence gene cluster (pVGC)

such as plcA (LMON_0199)[72], hly/LLO (LMON_0200)[73], mpl (LMON_0201)[72], actA (LMON_0202)[74], and plcB (LMON_0203)[72], as well as inlB (LMON_0442)[75], gap (LMON_2470)[76], and fbaA (LMON_2571)[77]. Due to the different HLA haplotypes of the two cell lines, not surprisingly there was limited overlap between both *Listeria* immunopeptide sets with only a single immunopeptide (HLPEFTNEV) from InlB presented on both cell lines. In contrast, substantially more overlap was evident at the antigen (protein) level with seven proteins represented on both cell lines, including several of the aforementioned pVGC virulence genes (Fig. 2f). Taken together, from two different infected cell lines we identified 68 MHC Class I presented peptides from 42 *Listeria* proteins, including several previously described antigens.

### *Listeria* antigens are often derived from the bacterial periphery
According to their predicted subcellular localization[78], the majority of the detected *Listeria* antigens is located either extracellularly or at the bacterial surface (Fig. 3a). This makes these antigens more easily accessible to the host antigen processing and presentation machinery, likely explaining their overrepresentation compared to bacterial cytoplasmic antigens, as suggested previously[55,56,79–81]. Similarly, the cluster of orthologous groups (COG) annotation revealed cell wall/membrane biogenesis as most common COG term for identified *Listeria* antigens (Fig. 3b). Mapping physical and functional associations between the identified *Listeria* antigens in the STRING database[82] yielded nine clusters of associated proteins, with the two largest clusters separating again peripheral (virulence) proteins from cytoplasmic proteins (Supplementary Fig. 4).

Interestingly, more than half of the identified *Listeria* peptides were derived from only thirteen bacterial proteins of which eleven were surface-exposed or secreted antigens (Fig. 3c). Such unequal presentation suggests immunodominance of these antigens, classifying them as potential vaccine candidates[57]. From these antigens, seven were identified on both cell lines and unsurprisingly comprised several well-known virulence factors including hly/LLO, Mpl, ActA, InlB, InlC, and PlcA. In addition, elongation factor Tu (EF-Tu) and glyceraldehyde-3-phosphate dehydrogenase (gap) were highly presented. Although EF-Tu and gap are abundant cytoplasmic bacterial proteins, alternative localization of these proteins to the bacterial periphery was recently described[83–85]. Interestingly, the antigen giving rise to most presented immunopeptides (seven on both cell lines combined) was the rather poorly characterized oligopeptide ABC transporter, periplasmic oligopeptide-binding protein OppA (TC 3.A.1.5.1, LMON_0149 in *Listeria monocytogenes* EGD). This protein is predicted to be involved in solute transport across the plasma membrane[86], similar to four other OppA proteins in *Listeria* of which two (LMON_2272 and LMON_0134) were also picked up in our screens (Fig. 3c). In conclusion, most detected *Listeria* immunopeptides were derived from antigens at the bacterial periphery. Highly presented antigens included major virulence factors, but also poorly characterized bacterial proteins without any known harmful activity to host cells that are therefore interesting vaccine candidates.

### mRNA vaccines encoding highly presented antigens provide prophylactic protection
To test whether highly presented antigens indeed provide high levels of protective immunity, seven *Listeria* proteins represented by two or more immunopeptides and with no known toxicity or enzymatic activity were selected as mRNA vaccine candidates, including LMON_0149, EF-Tu and LLO (depicted in green in Fig. 3c). Even though LLO naturally posseses toxicity as a pore-forming agent, we opted to select it as a vaccine candidate since it is denatured and rapidly degraded at (cytosolic) pH > 6[87]. Nevertheless, to further ensure safety of LLO as antigen we encoded the non pore-forming E262K LLO variant in our mRNA formulations[88]. Moreover, using the Vaxign2 vaccine design platform we in silico evaluated the selected vaccine candidates

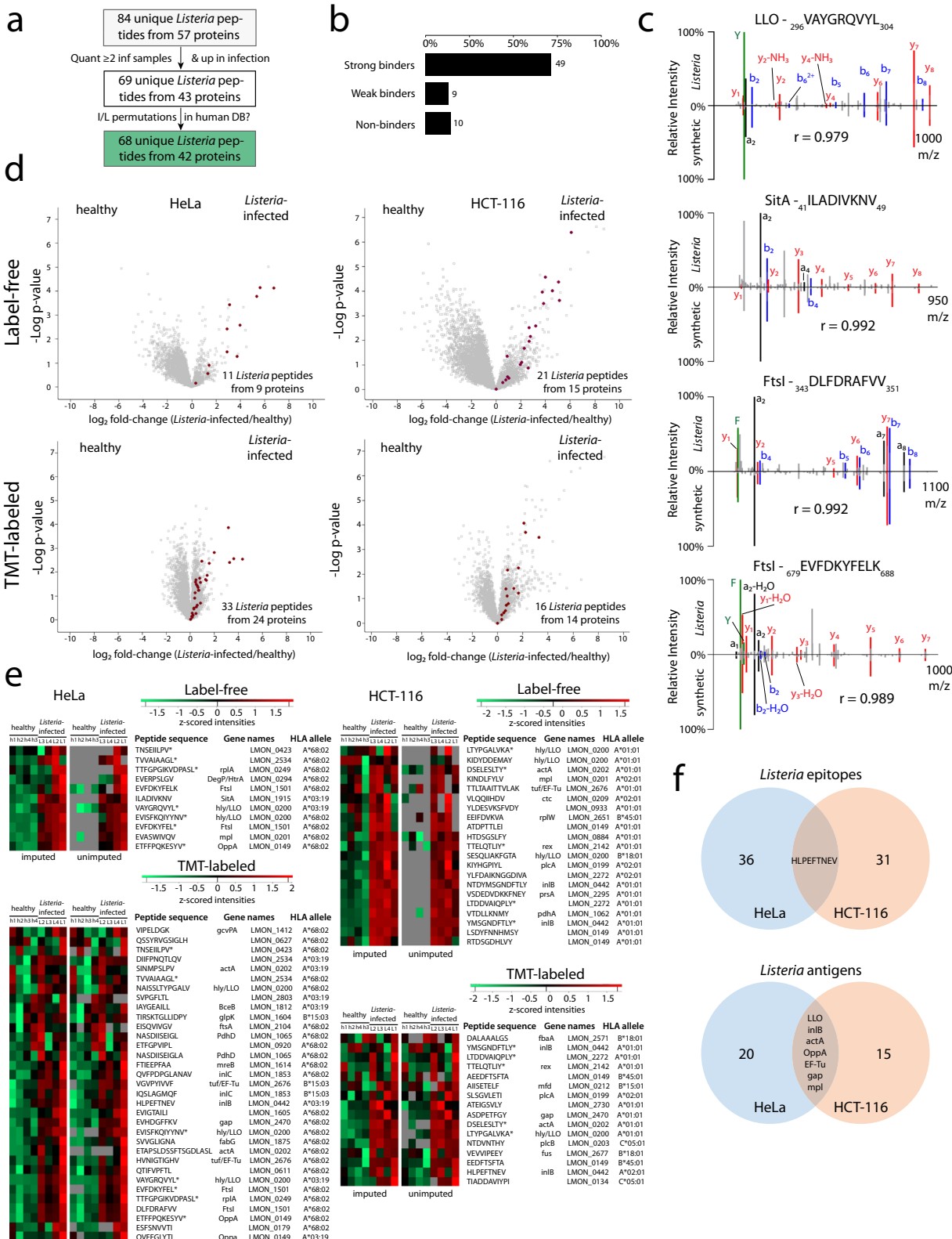

by means of reverse vaccinology, calculating Vaxign-ML scores for all candidates as well as all *Listeria monocytogenes* EGD proteins[89,90]. Vaxign-ML scoring is based on a machine learning algorithm utilizing nineteen antigen properties including immunogenicity, subcellular localization and number of transmembrane helices, amongst others[91]. Interestingly, the seven selected antigens showed superior average ranking and a significantly better average score when compared to all

42 detected antigens and to all 2847 *Listeria monocytogenes* EGD proteins (Fig. 3d, e).

We next tested the protection against *Listeria* inferred by the selected vaccine candidates in two prophylactic vaccination experiments. The seven vaccine candidates were encoded in in vitro transcribed N1-methylpseudouridine (m1Ψ) modified mRNA, which was complexed within cationic liposomes and administered as a vaccine

**Fig. 2 | Detection of high confident *Listeria* immunopeptides. a** Filtering of *Listeria* peptides for detection in at least two infected samples, higher overall abundance in infected samples and absence in human proteins resulted in 68 high confident bacterial peptides from 42 *Listeria* protein antigens (Supplementary Data 2). **b** Peptide binding affinity prediction using the NetMHCpan EL 4.1 algorithm demonstrated that the large majority of the high confidence *Listeria* peptides are indeed predicted to bind to at least one HLA class I allele of the respective cell line[67,134–136]. **c** A selection of 24 *Listeria* immunopeptide sequences was synthesized to compare their synthetic and experimental fragmentation spectra, confirming the bona fide identification of the four peptides shown here and in Supplementary Fig. 3. The correlation coefficient r is shown for each *Listeria*-synthetic peptide pair (see methods). **d** Volcano plots show that the high confident *Listeria* immuno-peptides belong to the most upregulated immunopeptides upon infection in both cellular models. **e** Likewise, heat maps visualizing individual *Listeria* peptide intensities (z-scored) following label-free quantitation show that peptides are generally absent in the uninfected control samples, while TMT-labeling data demonstrates more low-level background intensities presumably due to ratio suppression[68,69]. Immunopeptides identified by both label-free and TMT workflows are indicated by an asterisk (*), including the previously described VAYGRQVYL epitope from LLO[137]. Two versions of each heat map are shown, with the left one indicating z-scored intensities after imputing empty values with low values around the detection limit (to allow t-testing). The heat maps on the right are identical, but show originally missing (unimputed) values in gray. **f** Venn diagrams showing the overlap between both cell lines in presented *Listeria* peptides (top) and their parent protein antigens (bottom). Only a single peptide (HLVPEFTNEV) was detected on both cell lines, while the overlap at the antigen level is substantially higher with seven *Listeria* proteins being presented by the two cellular models. Source data are provided as a Source Data file.

to C57BL/6J mice. To this end, we made use of the galsome platform in which antigen-encoding nucleoside-modified mRNA is co-delivered in cationic liposomes together with the immunopo-tentiator α-galactosylceramide (α-GC) as an adjuvant[92]. Co-formulation of low amounts of the glycolipid α-GC was shown to activate invariant natural killer T (iNKT) cells resulting in elevated levels of antigen-specific cellular responses facilitating a potent adaptive immune response[28,92]. In both experiments, an ovalbumin-encoding mRNA (OVA) was included as negative control and inter-experiment reference. To test whether administration of a multi-antigen vaccine has the potential to yield higher levels of protection compared to single antigens (as recently demonstrated for SARS-Cov-2[93,94]), in the first experiment we added a combination vaccine in which mRNA encoding LLO_E262K was combined with mRNA encoding LMON_2272 in a single formulation. In the second experi-ment, we included an additional PBS negative control to elucidate the potential immunestimulatory effect of the galsome without pathogen-related antigens. Moreover, in this experiment, we also included a positive control injecting low amounts of *Listeria mono-cytogenes* EGD (1 × 10^4 bacteria) instead of an mRNA vaccine. These low-dose infections result in an acute listeriosis that can easily be overcome by the animals, leading to a protective adaptive immune response indicating the maximum level of protection that could potentially be reached by vaccination. In both experiments, prime vaccination on day 0 was followed by an identical booster on day 14 and *Listeria* infection on day 28. Three days post-challenge the ani-mals were sacrificed and the bacterial load in liver and spleen was assessed by counting colony-forming units (CFUs) (Fig. 4a).

Mice tolerated the vaccinations well with a maximal weight loss of 2.5% on the first day after vaccination and more than 75% of mice reached their full body weight again three days post-vaccination (Supplementary Data 3). All mice vaccinated with mRNA galsomes encoding *Listeria* antigens showed a lower bacterial burden in both spleen and liver in comparison to control vaccinations with ovalbumin (Fig. 4b and Supplementary Fig. 5A, B). Statistically significant reduc-tions were observed in both organs for LMON_0149, EF-Tu and the combination vaccine (LLO_E262K + LMON_2272). Vaccination with our best-presented antigen LMON_0149 resulted in a ~3 log CFU reduction in the spleen and a ~1.5 log reduction in the liver as compared to the OVA negative control, protection levels that were confirmed or even exceeded in two additional independent experiments (Supplementary Fig. 5C–E). Similarly, the EF-Tu mRNA vaccine diminished the bacterial CFUs by ~2 logs in the spleen and ~4 logs in the liver. While vaccina-tions with LLO_E262K or LMON_2272 alone did not significantly reduce the number of *Listeria*, the combined fomulation suppressed bacterial CFUs by ~1.5 log in spleen and ~3 logs in liver, suggesting that encoding multiple bacterial antigens can provide beneficial synergistic effects. These effects are however dependent on the particular antigen combination since combining LMON_0149 with LLO_E262K in an additional independent experiment did not lead to higher levels of protection compared to LMON_0149 alone (Supplementary Fig. 5E). Next to these best-performing antigens, vaccination with PdhD and inlB displayed significant levels of protection only in the liver with ~1 log CFU reductions for both antigens (Fig. 4b and Supplementary Fig. 5A, B). The sublethal *Listeria* infection as positive control could reduce the bacterial CFUs by 3.5 and 2.5 logs in spleen and liver, respectively, confirming the expected high levels of protection. Both the OVA and the PBS negative controls resulted in comparable high bacterial counts after infection in both liver and spleen, suggesting that the utilized galsome platform alone does not infer protection by itself, but only upon administration of pathogen-specific antigen mRNA. Interestingly, when considering all seven tested antigens we observed positive correlations (Pearson and Spearman's rank r values between 0.56 and 0.69) between the number of identified immuno-peptides/antigen and the percentage of CFU reduction, suggesting that the number of immunopeptides identified in immunopepti-domics experiments can indeed be used to prioritize bacterial vaccine candidates (Fig. 4c, d).

## mRNA vaccination with LMON_0149 induces specific CD8⁺ T-cell responses

Since protective immunity against *Listeria* mainly depends on T-cell-mediated immunity[28], we next tested whether mRNA vaccination with our best-presented antigen LMON_0149 induced specific CD8⁺ T-cell responses. To this end, mice were vaccinated with mRNA galsomes encoding LMON_0149 or OVA as control. After 7 days, splenocytes were isolated and pulsed with two synthetic peptide epitopes pre-dicted from the LMON_0149 sequence by the IEDB analysis resource tools NetMHCpan v4.1[67] and MHC-NP[95]. YSYKFIRV was tested as best predicted epitope binding to the MHC Class I H2-Kb allele expressed by C57BL/6J mice (Supplementary Data 4). We also included QVFE-GLYTL as a strong predicted binder for both H2-Db and H2-Kb since it is identical to one of the human LMON_0149 immunopeptides that we picked up from HeLa cells (Fig. 2e and Supplementary Data 2 and 4). Mice vaccinated with LMON_0149 showed detectable levels of IFN-y producing CD8⁺ T-cells to both YSYKFIRV and QVFEGLYTL, but not to SIINFEKL, a well-characterized OVA epitope (Fig. 5a). In contrast, mice vaccinated with OVA strongly responded to SIINFEKL but not to YSYKFIRV and QVFEGLYTL (Fig. 5b and Supplementary Fig. 6A), demonstrating that mRNA vaccination with LMON_0149 induces spe-cific CD8⁺ T-cell responses against this *Listeria* antigen.

Together, our data show that encoding highly presented *Listeria* antigens in mRNA vaccine formulations results in specific T-cell responses and high levels of protection in vaccination-challenge experiments in mice, indicating that immunopeptidomics holds great promise to discover novel bacterial vaccine candidates. The results presented in this study could be used to develop an effective mRNA vaccine against human or animal listeriosis and serve as a template to develop mRNA vaccines against other intracellular bacterial pathogens.

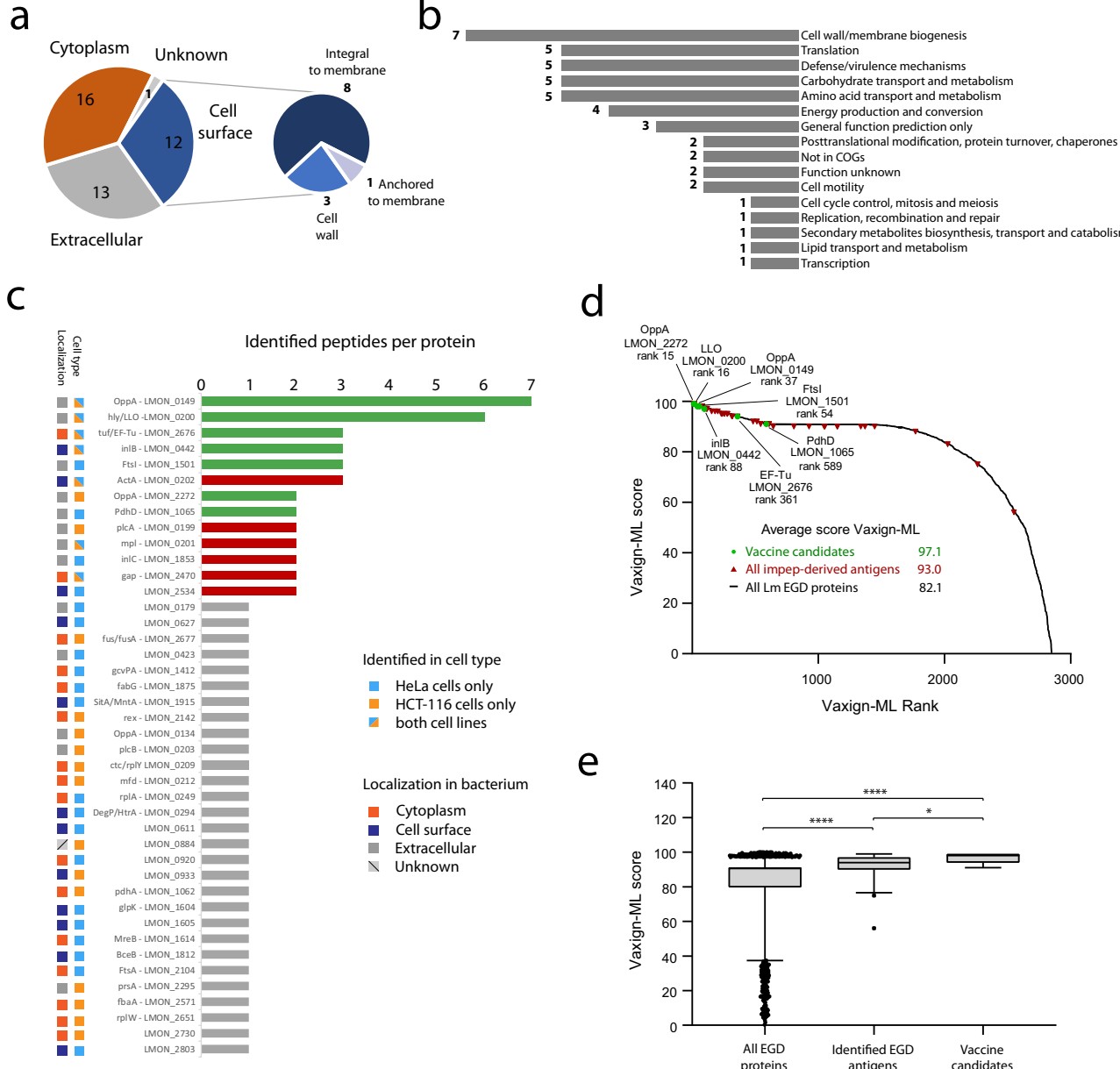

**Fig. 3 | *Listeria* antigens are often derived from the bacterial periphery.**
**a** Subcellular localization prediction of *Listeria* antigens indicated that the majority of antigens are localized at the bacterial periphery as either cell surface-associated or extracellular proteins[78]. **b** Clusters of Orthologous Groups (COG) terms assessment corroborated the importance of cell wall and membrane-associated proteins for presentation as antigens[138,139]. **c** Histogram showing the number of identified immunopeptides for all 42 detected *Listeria* antigens (Supplementary Data 2). The seven most presented antigens without any known enzymatic or harmful activity to host cells were selected as vaccination targets (green bars). Antigens with multiple peptides identified but known enzymatic or toxic properties were excluded from further assessment as vaccine candidates (red bars). Most of these selected antigens were identified from both cell types and are predicted to be present at the bacterial surface. **d** Vaxign-ML scores were calculated for all 2847 *Listeria*

*monocytogenes* EGD proteins and plotted according to scoring rank[90]. The seven selected *Listeria* antigens were among the top scoring proteins, further supporting their selection as vaccine candidates. **e** Box plot showing the Vaxign-ML scores for all 2847 EGD proteins, the 42 identified *Listeria* antigens and the seven selected antigens. The latter showed the highest average score, followed by all identified *Listeria* antigens, both scoring significantly higher than the average score of all EGD proteins (two-sided Mann-Whitney nonparametrical testing, $n = 2847$ scores for all EGD protein, $n = 42$ scores for identified antigens, $n = 7$ scores for selected vaccine candidates). Box plots visualize the median, with the box bounds showing the 25th and 75th percentiles and the whiskers the 5th and 95th percentile. Values below the 5th and above the 95th percentile are visualized as individual data points. Asterisks indicate *p* values with $*p < 0.05$ and $****p < 0.0001$. Source data are provided as a Source Data file.

## Discussion

Identifying immunologically relevant antigens that are presented on host cell surfaces has been challenging for intracellular bacteria due to analytical limitations. Two decades ago, the first epitopes from intracellular bacteria were identified in an untargeted way using MS-based immunopeptidomics[52]. While initial studies only yielded a handful of pathogen-derived immunopeptides, technological advances now

allow the detection of dozens of MHC-bound bacterial peptides presented on infected cells, recently reviewed in ref. 57. Translation of these MS-identified immunopeptides into safe, broadly applicable and effective vaccines is however lagging behind, in part due to the long development times for classical vaccines often using inactivated or attenuated pathogens[96]. We here present a workflow for the immunopeptidomics-based development of mRNA-based vaccines

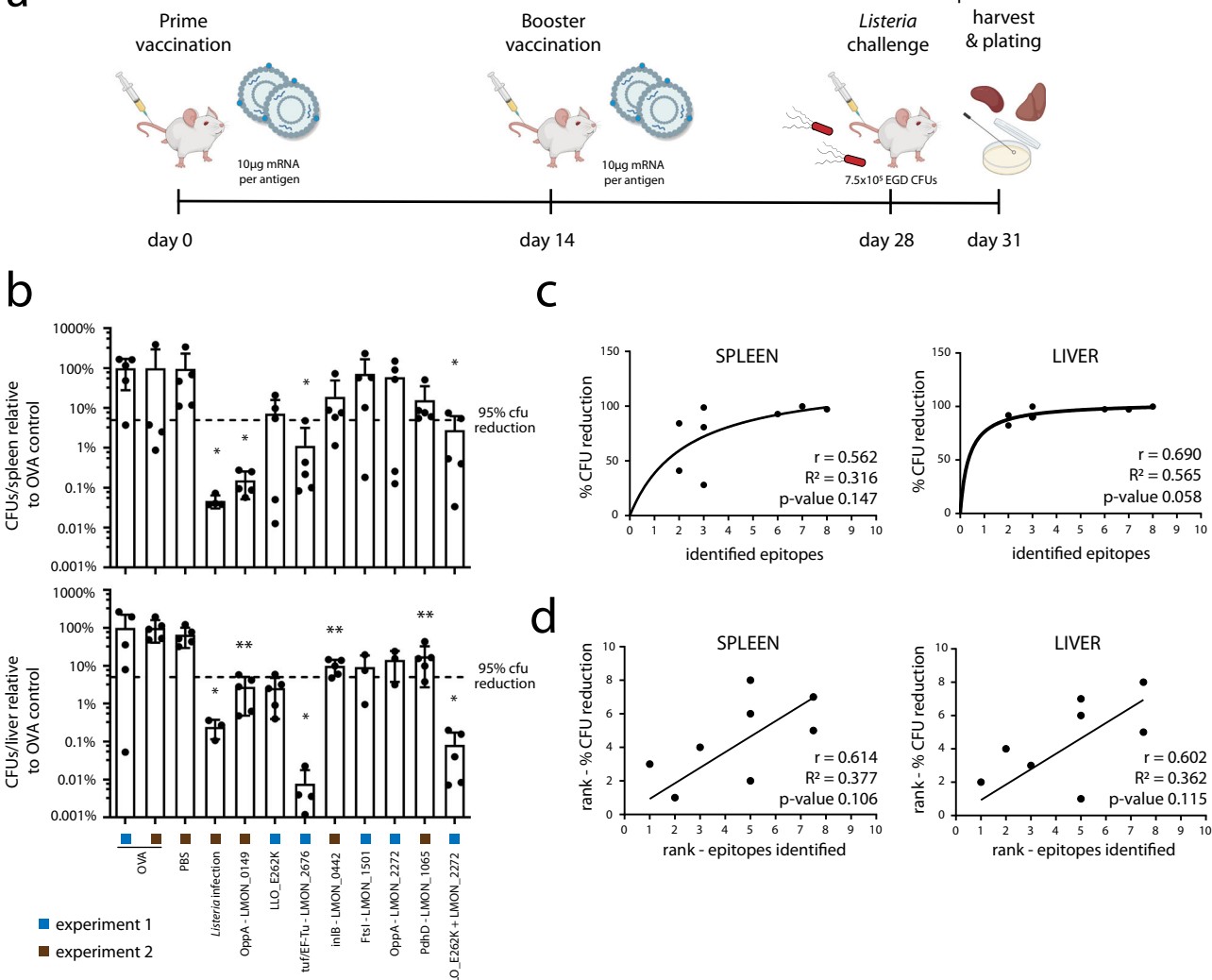

**Fig. 4 | Highly presented antigens provide protection as mRNA vaccine candidates. a** C57BL/6J mice were vaccinated utilizing full length, mRNA-encoded *Listeria* antigens formulated in α-GC adjuvanted cationic LNPs. All seven antigen candidates were each tested in one independent experiment (experiments 1 and 2). Additional experiments for LMON_0149 and LLO_E262K are shown in Supplementary Fig. 5. Mice were vaccinated intravenously with cationic LNPs comprising 10 µg of *Listeria* antigen mRNA. As negative control and inter-experiment reference, 10 µg ovalbumin (OVA) mRNA was injected in both experiments. In experiment 1, a combination vaccine was administered containing 5 µg LLO_E262K and 5 µg LMON_2272 mRNA. In experiment 2, PBS injection was included as additional negative control, while low-dose *Listeria monocytogenes* EGD infection ($1 \times 10^4$ CFUs) served as positive control. Two weeks after prime vaccination, an identical booster was administered and 2 weeks later animals were challenged by i.v. injection of $7.5 \times 10^5$ bacteria. Mice were euthanized 72 h post-challenge and bacterial loads in spleen and liver were assessed by CFU counting. Created with BioRender.com. **b** Bar charts depicting CFU counts in spleen (upper) and liver (lower)

relative to the OVA negative control. All *Listeria* vaccines reduced the bacterial burden, while only LMON_0149, EF-Tu and the combination vaccine reached statistical significance in both organs (representative results from single experiments, two-tailed Mann-Whitney test, data are presented as mean values ± SD, $n = 5$ individual animals except in liver for LMON_2272 ($n = 3$), FtsI ($n = 3$), EF-Tu ($n = 4$), *Listeria* infection ($n = 3$), and in spleen for OVA ($n = 4$) and *Listeria* infection ($n = 4$), where plating of the excluded replicates did not yield CFUs). Pearson (**c**) and Spearman rank (**d**) correlations were calculated with GraphPad Prism 9.3 between the number of identified bacterial immunopeptides per vaccine candidate and vaccination efficacy expressed as % CFU reduction. For the combination vaccine, peptide numbers for both antigens were summed up. In both liver and spleen, a positive correlation between the number of presented immunopeptides and protective efficacy is indicated by positive *r* values, although without reaching statistical significance. Asterisks indicate *p* values with *$p$ value < 0.05 and **$p$ value < 0.01. Source data are provided as a Source Data file.

against intracellular bacteria. We used *Listeria monocytogenes* as a clinically relevant bacterial model pathogen to infect HeLa and HCT-116 cells, two human epithelial cell lines, and we identified *Listeria* immunopeptides presented on MHC class I molecules by a hybrid MS approach, combining label-free and TMT-labeled measurements. Limited overlap of peptide identifications from label-free and TMT-labeling analyses suggests high orthogonality between the two different methods facilitating highly comprehensive immunopeptidomics screening. HeLa cells have been extensively used as an infection model in *Listeria* research and refined infection protocols are

available[97–99]. In addition, HeLa cells possess limited HLA allele diversity due to loss of heterozygosity leading to the expression of only three different HLA alleles aiding in immunopeptide analysis due to reduced immunopeptide complexity. Contrastingly, HCT-116 cells have been used rarely in *Listeria* research but were chosen for their intestinal origin and epithelial morphology mimicking natural *Listeria*-targeted cells[100]. HCT-116 cells furthermore possess an HLA haplotype comprising many common HLA alleles such as HLA-A*02:01, HLA-A*01:01, HLA-C*05:01 and HLA-C*07:01[101]. In contrast to antigen-presenting cells of myeloid origin, HeLa and HCT-116 cells only

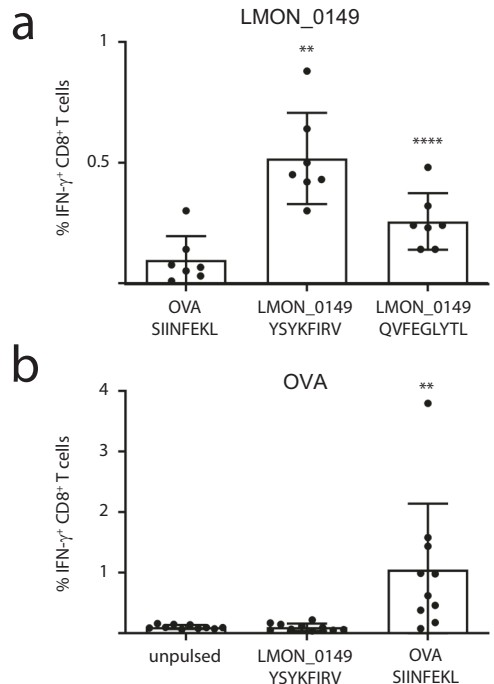

**Fig. 5 | Specific CD8⁺ T-cell responses upon LMON_0149 vaccination.** C57BL/6J mice were vaccinated with mRNA galsomes encoding **a** LMON_0149 or **b** OVA as control (10 mice/group). After 7 days, splenocytes were isolated and pulsed with a control OVA epitope or two synthetic peptide epitopes predicted from the LMON_0149 sequence using the IEDB resource tools NetMHCpan v4.1[67] and MHC-NP[95] (Supplementary Data 4). **a** Mice vaccinated with LMON_0149 showed significantly higher levels of CD8⁺ T-cell responses to both LMON_0149 YSYKFIRV and QVFEGLYTL epitopes as compared to the OVA epitope SIINFEKL (representative results of a single experiment, Shapiro-Wilk test confirmed data normality, paired two-sided *t*-test applied, data are presented as mean values ± SD, *n* = 7 individual animals since splenocytes from three animals with <35% cell viability were excluded). **b** Conversely, OVA-vaccinated mice responded well to SIINFEKL but not to YSYKFIRV, confirming that mRNA vaccination with LMON_0149 elicits specific CD8⁺ T-cell responses against this *Listeria* antigen (representative results of a single experiment, Shapiro-Wilk test rejected data normality, Wilcoxon matched-pairs signed rank test applied, data are presented as mean values ± SD, *n* = 10 individual animals). Additional control experiments with OVA-vaccinated or PBS-injected mice are shown in Supplementary Fig. 6. Asterisks indicate *p* values with **\*\*p* < 0.01 and \*\*\*\*\**p* < 0.01. Source data are provided as a Source Data file.

express negligible levels of MHC class II[63,66]. This strongly reduced the risk of accidental co-enrichment of MHC class II ligands in our screens, although some trace contamination cannot be excluded[102]. Nevertheless, in conjunction with serial dual step immunoprecipitation strategies[56,103–105], myeloid cells could be included in future screens to complement the MHC Class I immunopeptides reported here with MHC Class II-presented *Listeria* peptides.

We detected 68 highly confident peptides from 42 different *Listeria* antigens including several well-known virulence factors such as LLO, InlB, and ActA, as well as previously uncharacterized proteins. One example of the latter is LMON_0149 denoted as Oligopeptide ABC transporter, periplasmic oligopeptide-binding protein OppA, which we identified as most presented antigen with seven different immunopeptides detected on both cell lines. In *Listeria monocytogenes* EGD, five proteins are annotated as OppA (LMON_0149, LMON_2272, LMON_2584, LMON_0134, LMON_2115 in EGD), of which three, LMON_0149, LMON_2272 and LMON_0134, were represented by MHC Class I peptides in our data. The homolog of LMON_2272 in *Listeria monocytogenes* EDGe (lmo0152) has been demonstrated to mediate oligopeptide transport and facilitate bacterial growth at low temperatures, while it also plays a role in intracellular and in vivo

infection[106]. It is also involved in quorum sensing, beta-lactam resistance and acts as an ABC transporter according to the Kyoto Encyclopedia of Genes and Genomes[107–109]. The other four OppA proteins are less well studied, but based on structural similarities are likely also involved in solute transport across the plasma membrane[86] and regulated upon growth at low temperatures or in organs or blood[110].

The parent *Listeria* antigen proteins were ranked according to the identified peptide count and the top seven candidates devoid of any known enzymatic activity were selected as vaccine candidates. The antigen sequence information was translated into nucleoside-modified mRNA and formulated in cationic LNPs including the glycolipid α-GC as a smart adjuvant termed galsomes. This mRNA platform facilitates a pluripotent innate and adaptive immune response spearheaded by iNKT cells[92]. Upon vaccination, galsomes are taken up by dendritic cells (DCs) leading to surface presentation of (i) the mRNA-encoded antigens on MHC I molecules activating CD8⁺ T cells and (ii) α-GC on CD1d glycolipid receptors stimulating iNKT cells. iNKT cells can further activate DCs through CD40 ligation which enhances DC maturation thereby improving T cell activation capacity[111]. In addition, the response of iNKT cells to α-GC results in a burst release of cytokines and chemokines, which further promotes and regulates immunity[112]. Moreover, prior reports evidenced that activated NKT cells provide early protection against enteric *Listeria* infection with systemic production of IFN-γ[113].

Following prime-boost vaccination and *Listeria* challenge in mice, most antigens significantly reduced the bacterial load in the liver, while LMON_0149 and EF-Tu also showed significant protection in the spleen. Mice vaccinated with our top candidate LMON_0149 contained 1000× less bacteria in the spleen compared to OVA-vaccinated control animals. This reduction is less pronounced compared to reductions reported for live attenuated strains[27], but differences in vaccination schemes, *Listeria* strains used and infection doses make direct comparisons difficult. Most importantly, LMON_0149 vaccination resulted in protection that was almost as high as the protection offered by a low-dose *Listeria* infection. Given the short interval between infection and organ harvest, we suspect that LMON_0149 vaccinations would facilitate complete clearance given sufficient time for the immune system to completely eliminate the pathogen from the host. The specific CD8⁺ T-cell responses that we measured against two epitopes from LMON_0149 further indicated that vaccination induces cellular immunity, known to be required for protection against *Listeria*[28]. EF-Tu was a somewhat surprising hit as it is typically denoted as a highly abundant cytoplasmic protein functioning as elongation factor during protein biosynthesis and was therefore not expected to be readily available for host antigen processing[78,84]. However, more recent reports showcase a moonlighting function of EF-Tu as fibronectin-binding molecule at the cell surface[83,114,115], likely explaining its favorable results as a vaccine candidate and also its immunogenicity during infection with *Borrelia burgdorferi*, *Chlamydia trachomatis* and *Helicobacter pylori*[116–118]. While EF-Tu seems a promising vaccination candidate in *Listeria* offering a remarkably high (4 log) protection in the liver, it is highly conserved amongst prokaryotic species and might therefore potentially lead to autoimmune-like side effects against the host microbiome[119]. Strikingly, the combination of the two antigens LLO_E262K and LMON_2272 resulted also in statistically significant and high levels of protection in both spleen and liver similar to LMON_0149, while the individual, separate vaccinations did not achieve significant levels of protection. This hints toward a synergistic effect of this particular combination of antigens, a promising observation that warrants future testing of other antigen combinations including top candidates such as LMON_0149. An initial attempt to combine LMON_0149 vaccination with LLO_E262K did however not result in higher levels of protection compared to vaccination with LMON_0149 alone, indicating that effective combinations will be antigen-dependent. This result further highlights the surprisingly low

protection offered by LLO_E262K, in contrast with many mice and human trials where epitopes within LLO have been identified as immunodominant[39,40,73]. We speculate that this might be related to LLO's expression as a cytosolic intracellular protein in the case of mRNA vaccination versus delivery as an extracellular antigen in the case of vaccination with a recombinant subunit or live attenuated strain. At cytosolic pH >6 LLO is denatured and rapidly degraded[87], a process that might occur with different kinetics and that may result in different epitopes for host-cell expressed LLO versus bacterially expressed or ectopically delivered LLO.

To the best of our knowledge, this is the first report describing an mRNA-based, cell-free vaccine against an intracellular bacterium demonstrating high levels of protection in vaccination-challenge experiments. Previous LNP mRNA vaccines against bacterial pathogens such as *Mycobacterium tuberculosis* and Group A and B *Streptococci* showed promising results, but either did not report, or reported only moderate reductions in bacterial burden upon vaccination[46–48,120]. Recently, an mRNA vaccine encoding nineteen saliva proteins from *Ixodes scapularis* (black-legged tick) led to acquired tick resistance and reduced transmission of the Lyme disease agent *Borrelia burgdorferi* in Guinea pigs, however, in this case the vector instead of the pathogen was targeted by the vaccine[121]. It is noteworthy that despite their initial detection on infected human cells, the *Listeria* antigens identified here are also immunologically relevant for the chosen mouse model as evidenced by the vaccination-challenge experiments. These antigens, therefore, hold great promise for further development of *Listeria* vaccines for both human and animal use. Preclinical studies in humanized mice will help to determine the protective potential in man. Moreover, high conservation levels of the identified immunopeptides suggest that the current *Listeria monocytogenes* EGD-based sequences encoded in the mRNA will provide protection against a broad range of *Listeria monocytogenes* strains, including clinical and veterinary isolates (Supplementary Fig. 7).

Finally, beyond the development of an anti-*Listeria* vaccine, the encountered correlation of highly presented antigens inferring the greatest levels of protection could facilitate the process of vaccine antigen prioritization, speeding up vaccine development and preserving valuable resources in the battle against rising antimicrobial resistance (AMR) levels of bacterial pathogens. Future screens will show whether this hypothesis holds true. In this regard, the present study can act as a blueprint for immunopeptidomics-based development of mRNA vaccines against intracellular bacterial pathogens. Moreover, the *Listeria* antigens and epitopes identified here could be used to improve *Listeria* strains that are tested as cancer vaccine vectors. Due to the preferential presentation of immunodominant *Listeria* epitopes, these vectors were recently reported to suffer from reduced efficacy in cargo antigen presentation[42]. Future efforts could attempt to delete several of the novel *Listeria* antigens reported here or to mutate their epitope anchor residues in order to abolish MHC binding and to free up presentation capacity for the actual target cancer epitopes.

## Methods

The authors confirm that all animal experiments were done under conditions specified by law and authorized by the UGent Institutional Ethical Committee on Experimental Animals. The animal facility, located at the VIB-UGent Center for Inflammation Research, Ghent, Belgium, operates under the Flemish Government License Number LA1400536.

### Cell culture

Human HeLa cells (ECACC 93021013), HCT-116 cells (ECACC 91091005), and JY cells (ECACC 94022533) were cultured at 37 °C in a humidified atmosphere at 10% $CO_2$. HeLa cells were grown without antibiotics in MEM medium (#M2279, Merck) supplemented with 10% fetal bovine serum (FBS, #10270106, Thermo Fisher Scientific), 2 mM GlutaMax (#35050038, Thermo Fisher Scientific), 1% non-essential amino acids (#11140035, Thermo Fisher Scientific), 1 mM sodium pyruvate (#11360039, Thermo Fisher Scientific) and 10 mM HEPES (#15630056, Thermo Fisher Scientific). HCT-116 cells were maintained without antibiotics in McCoy's 5 A (modified) medium, HEPES (#22330070, Thermo Fisher Scientific) supplemented with 10% FBS and 2 mM GlutaMax. JY cells were cultured without antibiotics in RPMI medium (#61870036, Thermo Fisher Scientific) supplemented with 10% FBS and 2 mM GlutaMax. HeLa and HCT-116 cells were passaged at around 75% confluency in T175 flasks (#660160, Greiner Bio-One) and all three cell lines were tested and confirmed negative for mycoplasma contamination.

### *Listeria* infection of HeLa and HCT-116 cells

*Listeria monocytogenes* EGD (BUG600 strain) was grown in brain heart infusion (BHI) broth (#10462498, Fisher Scientific) shaking at 37 °C. *Listeria* was cultured overnight and then subcultured 1:20. At a density of 1E9/ml, bacteria were washed three times with PBS (#14040-174, Life Technologies) and resuspended in HeLa or HCT-116 growth medium without FBS prior to infection at an MOI of 50. For infection, HeLa or HCT-116 cells were grown in T175 flasks to a density of 7E6 cells/flask (HeLa) and 15E6 cells/flask (HCT-116). Directly before infection, cells were washed with PBS and 20 mL bacterial inoculum was added followed by incubation for 1 h at 37 °C and 10% $CO_2$ to allow bacterial entry. Afterwards, cells were washed two times with PBS and then grown further for 23 h in cell culture medium with 10% FBS, supplemented with 40 μg/mL of gentamicin (#G1397-10ML, Sigma-Aldrich, Merck) to kill extracellular bacteria. Cells were harvested using cell dissociation buffer (#13151-014, Life Technologies), washed three times with PBS and the dry cell pellet was stored at −80 °C until further processing.

### Generation of immunoaffinity columns for MHC Class I pull down

W6/32 antibody was purified from hybridoma cell (HB-95™, ATCC) supernatant as recommended by the cell line provider. To generate immunoaffinity columns[122], 0.5 mL of precipitated protein A sepharose 4B beads (#101041, Thermo Fisher Scientific) were washed with 100 mM Tris pH 8.0 (#0210313305, MP Biomedicals) before 3 mg of purified W6/32 antibody was added and allowed to bind at room temperature for 1 h in a rolling tube. W6/32-bound sepharose beads were then washed with 0.2 M sodium borate pH 9 (#B3545-500G, Sigma-Aldrich, Merck) and fresh 20 mM dimethylpimelimidate (#D8388-250MG, Sigma-Aldrich, Merck) dissolved in sodium borate solution was added for cross-linking of the W6/32 antibody to the beads. Cross-linking occured for 30 min in a rolling tube after which beads were washed with 0.2 M ethanol amine pH 8 (#149582500, Thermo Fisher Scientific) to quench the crosslinking reaction.

### Isolation and purification of immunopeptides

Cells were lysed by addition of a mild lysis buffer containing 1% octyl-β,D-glucopyranoside (#O9882-500MG, Sigma-Aldrich, Merck), 0.25% sodium deoxycholate (#1065040250, Millipore, Merck), 1.25x cOmplete protease inhibitor cocktail (#4693159001, Roche), 1 mM phenylmethylsulfonyl fluoride (#52332-5GM, Sigma-Aldrich, Merck), 0.2 mM iodoacetamide (#I1149-5G, Sigma-Aldrich, Merck) and 1 mM ethylendiamine tetraacetic acid (#EDS-100G, Sigma-Aldrich, Merck) in Ca/Mg-free PBS (#14190-169, Thermo Fisher Scientific). Ice cold lysis buffer was added at a ratio of 1 mL per 1E8 cells and lysis occured for 1 h on ice facilitated by vortexing and pipetting up and down the lysate every 5 min. Lysates were then cleared by initial centrifugation at 2000 × $g$ for 10 min at 4 °C, and supernatants were further cleared at 16,100 × $g$ for 35 min at 4 °C. Prior to immunoprecipitation, W6/32 immunoaffinity columns were washed with 0.1 M acetic acid (#1000562500, Sigma-Aldrich, Merck), followed by 100 mM TRIS pH 8.

Supernatants were added to the washed W6/32 immunoaffinity columns and precipitated overnight while rolling at 30 rpm at 4 °C. Reusable Econo glass columns (#7374150, Bio-Rad) were used for the immunoprecipitation. Beads were washed with ice-cold solutions in the cold room: twice with 150 mM sodium chloride (#S0520, Duchefa Biochemie) in 20 mM TRIS pH 8, twice with 400 mM NaCl in 20 mM TRIS pH 8, again twice with 150 mM NaCl in 20 mM TRIS pH 8 and finally twice with 20 mM TRIS pH 8. MHC Class I: peptide complexes were eluted by applying 5 mL 10% acetic acid per 500 μL settled beads.

The eluate was further acidified to a final concentration of 0.5% trifluoroacetic acid (#85183, Thermo Fisher Scientific) and pH was checked to be at 2.5 or below, prior to loading on preconditioned C18 ODS 100 mg SampliQ columns (#5982-1111, Agilent Technologies) using a vacuum manifold. After initial loading, samples were re-loaded four times before washing with 1 mL of 2% acetonitrile (ACN) (#1000292500, Sigma-Aldrich, Merck) in 0.2% acetic acid. Next, MHC class I peptides were specifically eluted by applying twice 300 μL of 30% acetonitrile in 0.2% trifluoracetic acid, followed by pooling of the eluates and complete drying in 2 mL protein LoBind tubes (#0030108450, Eppendorf). For further purification, immunopeptides were reconstituted in 100 μl of 2% ACN in 0.2% TFA for 15 min in an ultrasonic bath. OMIX C18 pipette tips (#A57003MB, Agilent Technologies) were conditioned three times with 200 μl of 80% ACN in 0.2% TFA, followed by five times 200 μl of 0.2% TFA. Resolubilized MHC-peptides were loaded onto the conditioned OMIX tips by pipetting up and down ten times, washed with 100 μL of 0.2% TFA and eluted by pipetting up and down ten times with 80 μl of 30% ACN in 0.2% TFA, followed by 20 μl of 30% ACN in 0.2% TFA. Eluates were pooled and divided into two equal fractions per sample to allow parallel label-free and TMT-labeling analysis. Both aliquots were completely dried and stored at −20 °C until further use.

## TMT labeling and pre-fractionation of immunopeptides

Dried immunopeptides were dissolved in 10 μl of 100 mM tetraethylammonium bicarbonate (#T7408-100ML, Sigma-Aldrich, Merck) by vortexing and sonicating for 15 min. TMT10plex labels (#90110, Thermo Fisher Scientific) were dissolved in 41 μl of anhydrous ACN and were regularly vortexed for 5 min to completely dissolve the labels. Next, 4.1 μl of TMT-label was added to each sample of peptides. For HeLa, uninfected samples 1 to 4 were labeled with the 127N, 127C, 128N and 128C TMT labels, while *Listeria* infected samples were labeled with 129N, 129C, 130N, and 130C. For HCT-116, uninfected samples 1 to 4 were labeled with 126, 127N, 127C and 128N, while *Listeria* infected samples were labeled with the 129C, 130N, 130C and 131 TMT reagents. Peptides were incubated with the TMT-labels for 1 h at room temperature while shaking at 700 rpm. 1 μl of hydroxylamine (#15675820, Fluka, Thermo Fisher Scientific) was then added to quench the reaction followed by incubation for 15 min at room temperature while shaking at 700 rpm. After quenching, the TMT-labeled samples from each cell line were pooled and dried completely. The TMT-labeled and pooled immunopeptides were then separated into 12 fractions using a reversed-phase C18-column at pH 10 and pH 5.5 for HeLa and HCT-116 cells, respectively. Dried peptides were solubilized in 100 μl of 2% ACN and 0.1% TFA in ultrapure water. 95 μl thereof was injected into an LC-system, consisting of a capillary pump (#G1376A, Agilent), an isocratic pump (#G1310A, Agilent), a multiple wavelength detector (#G1365B, Agilent), a column compartment (#G1316A, Agilent), a degasser (#G1379B, Agilent) and a well-plate autosampler (#G1367A, Agilent). Peptides were first loaded onto a 4 cm trapping column (made in-house, 250 μm internal diameter, 5 μm beads diameter, C18 Reprosil-HD, Dr. Maisch, Germany) at a flow rate of 25 μl/min. As mobile phase, two different solvents were used. Solvent A consisted of 10 mM ammonium bicarbonate (#09830, Sigma-Aldrich, Merck) and 2% ACN in ultrapure water while solvent B consisted of 10 mM ammonium bicarbonate and 70% ACN in ultrapure water. The pre-fractionation

started with 0% B followed by a linear increase from 0 to 100% B in 100 min between minute 20 and 120. The gradient was followed by a stationary washing phase at 100% B for 5 min and re-equilibration with 0% B for 15 min. Eluting fractions were collected using a Probot micro-fraction collector (#161403, LC-packings) into 12 MS-vials. Fractions were collected every minute from minute 20 onwards. After the first 12 fractions were collected in vials 1 to 12, the 13th fraction was again collected in vial 1 to re-start the collection cycle and to pool fractions in a smart way ensuring homogenous distribution of peptide hydrophobicity within each MS vial. Fractions were collected for a total of 84 min and the fractionated samples were vacuum-dried and stored at −20 °C prior to LC-MS/MS analysis.

## LC-MS/MS and data analysis of immunopeptides

Purified immunopeptides for label-free analysis were redissolved in 15 μl loading solvent (0.1% trifluoroacetic acid (TFA) in water/acetonitrile (ACN) (98:2, v/v)) from which 10 μL was injected for LC-MS/MS analysis on an Ultimate 3000 RSLC nano-LC system (Thermo Fisher Scientific) in-line connected to a Q Exactive HF mass spectrometer (Thermo Fisher Scientific) equipped with a nanospray flex ion source (Thermo Fisher Scientific). Trapping was performed at 10 μl/min for 4 min in loading solvent on a 20-mm trapping column (made in-house, 100 μm internal diameter, 5 μm beads, C18 Reprosil-HD, Dr Maisch, Ammerbuch-Entringen, Germany). Peptide separation after trapping was performed on a 200 cm micropillar array column (μPAC, PharmaFluidics) with C18-endcapped functionality. The Ultimate 3000's column oven was set to 50 °C and for proper ionization, a fused silica PicoTip emitter (10 μm inner diameter, New Objective, Littleton, MA, US) was connected to the μPAC outlet union and a grounded connection was provided to this union. Peptides were eluted by a non-linear gradient from 1 to 55% MS solvent B (0.1% FA in water/acetonitrile (2:8, v/v)) over 145 min, starting at a flow rate of 750 nl/min switching to 300 nl/min after 25 min, followed by a 15-min washing phase plateauing at 99% MS solvent B. Re-equilibration with 99% MS solvent A (0.1% FA in water) was performed at 300 nl/min for 45 min followed by 5 min at 750 nl/min adding up to a total run length of 210 min. The mass spectrometer was operated in data-dependent, positive ionization mode, automatically switching between MS and MS/MS acquisition for the ten most abundant peaks in a given MS spectrum. The source voltage was 2.2 kV, and the capillary temperature was set at 275 °C. One MS1 scan (*m/z* 300–1650, AGC target $3 \times 10^6$ ions, maximum ion injection time 60 ms), acquired at a resolution of 60,000 (at 200 *m/z*), was followed by up to ten tandem MS scans (resolution 15,000 at 200 *m/z*) of the most intense ions fulfilling predefined selection criteria (AGC target $1 \times 10^5$ ions, maximum ion injection time 120 ms, isolation window 1.5 Da, fixed first mass 100 *m/z*, spectrum data type: centroid, intensity threshold $8.3 \times 10^3$, exclusion of unassigned, 4–8, >8 positively charged precursors, peptide match off, exclude isotopes on, dynamic exclusion time 12 s). The higher-energy collisional dissociation was set to 28% normalized collision energy, and the polydimethylcyclosiloxane background ion at 445.12003 Da was used for internal calibration (lock mass).

Fractionated and TMT-labeled immunopeptides were redissolved in 20 μl loading solvent from which 15 μL was injected for LC-MS/MS analysis on an Ultimate 3000 RSLC nano-LC system (Thermo Fisher Scientific) in-line connected to a Fusion Lumos mass spectrometer (Thermo Fisher Scientific). Trapping was performed as described above and peptides were again separated on a 200 cm-long micropillar array column (μPAC, PharmaFluidics) with C18-endcapped functionality. Peptides were eluted by a non-linear gradient from 1 to 55% MS solvent B over 87 min, starting at a flow rate of 750 nl/min switching to 300 nl/min after 15 min, followed by a 13-min washing phase plateauing at 99% MS solvent B. Re-equilibration with 99% MS solvent A was performed at 300 nl/min for 40 min adding up to a total run length of 140 min. The mass spectrometer was operated in data-

dependent, positive ionization mode, automatically switching between MS and MS/MS acquisition to enable a cycle time of 3 s. One MS1 scan ($m/z$ 300–1650, AGC target $4 \times 10^5$ ions, maximum ion injection time 50 ms), acquired at a resolution of 120,000 (at 200 $m/z$), was followed by tandem MS scans in the orbitrap (resolution 50,000 at 200 $m/z$) of the most intense ions fulfilling predefined selection criteria (AGC target $7.5 \times 10^4$ ions, maximum ion injection time 120 ms, isolation window 1 Da, fixed first mass 100 $m/z$, spectrum data type: centroid, intensity threshold $8.3 \times 10^3$, including, 2–5 positively charged precursors, peptide match off, exclude isotopes on, dynamic exclusion time 60 s). The higher-energy collisional dissociation was set to 38% normalized collision energy, and the polydimethylcyclosiloxane background ion at 445.12003 Da was used for internal calibration (lock mass).

Mass spectrometry raw data were searched with PEAKS Studio X + ™ (version 10.5 build 20200219, Bioinformatics Solutions Inc, Waterloo, Canada) against a database of the human sequences in UniProt SwissProt (version January 2019, 20,413 entries) merged with *Listeria monocytogenes* EGD sequences from TrEMBL (version April 2019, 2847 entries)[123]. Databases were merged using dbtoolkit 2.0 (version 4.2.5)[124]. The peptide length was restricted to 8–30 amino acids, and unspecific digestion was chosen as digest mode. Methionine oxidation and N-terminal acetylation were set as variable modifications, and mass error tolerances were set to 10 ppm and 0.02 Da for parent and fragment ions, respectively. For indicating potential contaminant peptides, the MaxQuant contaminant database (MQ version 1.6.3.4) was enabled[125]. FDR estimation was carried out using the decoy-fusion approach. Identified peptide sequences were filtered at the PSM level for an FDR of 1% or better prior to label-free or TMT-10plex quantification in PEAKS Studio. Quantification results were not filtered and TMT-labeled peptide quantifications were not normalized before export as csv files for further processing using the Perseus software platform[126]. Exported csv files for label-free and TMT-labeled data were loaded into Perseus separately and intensity values were $\log_2$ transformed. After categorical annotation into uninfected, healthy and *Listeria*-infected sample groups, the data were filtered for at least two valid values in at least one sample group. Missing values were imputed from a normal distribution around the detection limit and a principal component analysis was performed. Volcano plots were generated by plotting the results of a two-sided Student's $t$ test of *Listeria*-infected against uninfected samples employing permutation-based multiparameter correction at an FDR of 5%. Heat maps were constructed by z-scoring *Listeria*-derived peptide $\log_2$ intensities before hierarchical clustering.

## Calculation of correlation coefficients between *Listeria*-derived and synthetic peptides

Python 3.7 was used to calculate spectral correlation including spectrum_utils version 0.3.5 and pyteomics version 4.5.2[127,128]. Spectrum processing was performed by annotation of fragment ion peaks for a, b and y ions including singly and doubly charged ions, followed by removal of precursor peaks for up to two isotopes and removal of low intensity (<5% of the maximum) peaks. All steps used a 50 ppm mass error tolerance. Pearson correlations were calculated on the intensities of all annotated fragment ions per spectrum. The code including an example peptide can be found on GitHub (https://github.com/RalfG/2022-listeria-spectrum-similarity) and Zenodo (https://doi.org/10.5281/zenodo.5948475). A runnable version of the script can be found online at Binder (https://mybinder.org/v2/gh/RalfG/2022-listeria-spectrum-similarity/HEAD?labpath=2022-listeria-spectrum-similarity.ipynb).

## mRNA production and liposome vaccine formulation

The protein sequences of the selected seven *Listeria* genes were cloned into a pGEM4z-plasmid vector (Promega) containing a T7 promoter, 5′ and 3′ UTR of human β globulin, and a poly(A) tail by Genscript. The *Listeria monocytogenes* EGD protein sequences were retrieved from the Listeriomics platform[129], codon optimized for mouse using the IDT codon optimization tool, and the final plasmid product was confirmed by sequencing (Supplementary Table 1). For the IVT mRNA production, plasmids were linearized with PstI (New England Biolabs, MA, USA) and purified using a PCR purification kit (Roche, Upper Bavaria, Germany). Linearized plasmids were used as templates for the in vitro transcription reaction using the T7 MegaScript kit, including an Anti-Reverse Cap Analog (ARCA, Trilink BioTechnologies), and chemically modified N1-methylpseudouridine-5′-triphosphate (Trilink BioTechnologies) instead of the normal nucleotide, uridine. The resulting capped mRNAs were purified by DNase I digestion, precipitated with LiCl and washed with 70% ethanol. All mRNAs were analyzed by agarose gel electrophoresis and concentrations were determined by measuring the absorbance at 260 nm. mRNAs were stored in small aliquots at −80 °C at a concentration of 1 μg/μL.

The mRNA constructs encoding the different *Listeria* antigens were formulated in cationic liposomes containing the immunopotentiator α-galactosylceramide (α-GC), as described previously[92]. DOTAP (1,2-dioleoyl-3-trimethylammonium-propane), cholesterol, and α-GC were purchased from Avanti Polar Lipids (Alabaster, USA). Cationic liposomes of DOTAP-cholesterol (2:3 molar ratio) were prepared by the thin-film hydration method. The appropriate amounts of lipids, dissolved in chloroform were transferred into a round-bottom flask. For the incorporation of the glycolipid antigen, 0.015 mol % of the total lipid amount was replaced by α-GC. The chloroform was evaporated under nitrogen, after which the lipid film was rehydrated in HEPES buffer (20 mM, pH 7.4, Sigma-Aldrich) to obtain a final lipid concentration of 12.5 mM. The resulting cationic liposomes were sonicated in a bath sonicator (Branson Ultrasonics, Dansbury, CT, USA). Then, they were mixed with mRNA to obtain mRNA lipoplexes at a cationic lipid-to-mRNA (N/P) ratio of 3, in a final formulation of an isotonic HEPES buffer containing 5% glucose (Sigma-Aldrich). The cationic lipoplex formulations were subjected to size and zeta potential quality control using a Malvern Zetasizer nano-ZS (Malvern Instruments Ltd., Worcestershire, UK).

## Mouse housing, prime-boost vaccination, and *Listeria* infection

Female C57BL/6J mice were ordered from Charles River Laboratories, France. The animals were housed in a temperature- (21 °C) and humidity- (60%) controlled environment with 12 h light/dark cycles; food and water were provided ad libitum. The animal facility, located at the VIB-UGent Center for Inflammation Research, Ghent, Belgium, operates under the Flemish Government License Number LA1400536. All experiments were done under conditions specified by law and authorized by the UGent Institutional Ethical Committee on Experimental Animals. *Listeria monocytogenes* (EGD BUG600 strain) was grown in BHI medium at 37 °C. Bacteria were cultured overnight and then sub cultured 1:10 in BHI medium for 2 h at 37 °C. Bacteria were washed three times in PBS and resuspended in PBS at $7.5 \times 10^5$ bacteria per 100 μl (~3× the LD50 of EGD[130]) or further diluted to $10^4$ bacteria per 100 μl. Female C57BL/6J mice (Charles River Laboratories, France) at 7 weeks of age were vaccinated intravenously (i.v.) by tail vein injection with either mRNA galsomes (10 μg mRNA, total volume 100 μl in isotonic HEPES-buffered glucose solution), a sublethal dose of *Listeria monocytogenes* ($1 \times 10^4$ bacteria in 100 μl PBS), or PBS (100 μl) at day 0 and day 14 of the experiment. Combination vaccines were administered by mixing the respective ready-to-use mRNA galsomes 1 + 1 resulting in the administration of 5 μg mRNA for both antigens. Mouse body weight was monitored for 72 h post-vaccination to assess potential adverse vaccination reactions. On day 28, the mice were infected i.v. by tail vein injection with $7.5 \times 10^5$ bacteria per animal. Mice were sacrificed 72 h following infection. CFUs per organ (liver or spleen) were enumerated by serial dilutions and plating on BHI agar

after tissue dissociation in sterile saline. For tissue dissociation, cell strainers and PBS were employed for spleens, while livers were disintegrated using PBS with 2% tween.

## Measurement of T-cell responses

Female C57BL/6J mice (Charles River Laboratories, France) at 7 weeks of age were i.v. injected with galsomes containing mRNA encoding the LMON_0149 antigen or ovalbumin (10 µg mRNA, 10 mice/group) or with PBS (100 µl). On day 8, $2 \times 10^6$ splenocytes from each animal were transferred in a round bottom 96 well plate (200 µl volume) and ex vivo restimulated with 1 µg/ml of the LMON_0149 peptides YSYKFIRV (GenScript) and QVFEGLYTL (made in house by solid phase synthesis) or the OVA-derived peptide SIINFEKL (Eurogentec, Seraing, Belgium) as control in the presence of a protein transport inhibitor cocktail of Brefeldin A and Monensin (eBioscience). Following 37 °C incubation for 5 h, cells were stained with fixable viability dye Aqua (#L34965, Thermo Fisher Scientific), incubated with Fc block (CD16/32) to block nonspecific FcR binding TruStain FcX (1:200, #101320, Biolegend), and surface stained with antibody CD8a (53-6.7) APC (1:100, #100712, Biolegend) (Fig. 5) or with antibody CD8a (53-6.7) FITC (1:100, #100706, Biolegend) and TCRbeta (H57-597) APC (1:100, #17596182, Thermo Fisher Scientific) (Supplementary Fig. 6). Cells were then fixed and permeabilized with BDCytoFix/CytoPerm solution (#554714, BD Biosciences), intracellular staining using a IFN-γ (XMG1.2) PE antibody (1:50, #505808, Biolegend) was performed in Cytoperm buffer for 30 min at RT. After additional washing steps, samples were measured by a MACSQuant Analyzer 16 (Miltenyi Biotec) and analyzed with FlowJo® software (BD Biosciences). Three spleen samples from LMON_0149 vaccinated mice (Fig. 5) and two spleen samples from PBS vaccinated mice that showed cell viability lower than 35% were excluded from the analysis. The gating strategy is outlined in Supplementary Fig. 6.

## Monitoring MHC expression by western blotting and flow cytometry

HeLa and JY cells were grown in a six well-plate to a density of 0.5 E6 cells/well and infected with *Listeria* for 24 h at an MOI of 25, treated with 10 ng/mL interferon-γ (#11343536, Immunotools) for 48 h or left untreated. For western blotting, cells were lysed in 1× Laëmmli buffer containing 62.5 mM Tris-HCl pH 6.8, 2% SDS, 10% glycerol, 0,005% Bromophenol blue (#J63615, Alfa Aesar) supplemented with 20 mM DTT (#D0632, Merck). Protein samples were boiled for 10 min at 95 °C and sonicated prior to SDS-PAGE. Samples were loaded on 4–15% polyacrylamide gradient gels (#M41215, Genescript) according to the guidelines of the manufacturer. Proteins were transferred to PVDF membrane (#IPFL00010, Merck) for 30 min at 100 V with Tris/Boric buffer at 50 mM/50 mM. Membranes were blocked for 1 h at room temperature (RT) with blocking buffer (#927-50000, LI-COR) and incubated with primary antibodies overnight at 4 °C diluted to 1:1,000 in TBS. The following primary antibodies were used: anti-HLA-ABC (#15240-1-AP, Proteintech), anti-HLA-DM (#21704-1-AP, Proteintech), anti-HLA-DR (#15862-1-AP, Proteintech), anti-α-tubulin (#sc-5286, Santa Cruz Biotechnology), anti-LLO (#ab200538, Abcam) and anti-STAT1 (#sc-464, Santa Cruz Biotechnology). The next day, membranes were washed three times for 15 min with TBS-Tween 0.1% (v/v) buffer and further incubated at RT for 1 h with the appropriate secondary antibody diluted to 1:5000 (anti-mouse # 926-32210 or anti-rabbit # 926-32211, Li-COR). Membranes were washed twice with TBS-tween 0.1% (TBS-T) and once with TBS prior to detection. Immunoreactive bands were visualized on a LI-COR-Odyssey infrared scanner (Li-COR). For flow cytometry analysis, cells were first stained with fixable viability dye Zombie green (#423111, Biolegend) and then incubated with Fc block TruStain FcX (CD16/32) to block nonspecific FcR binding (#422302, Biolegend). To detect MHC-I and MHC-II, cells were stained with antibodies against HLA-ABC (W6/32) PE (1:100, #311405, Biolegend) and HLA-DR/DP/DQ (Tü39)-APC (1:100, #361713, Biolegend),

respectively. Cells were fixed with 4% PFA for 40 min (#15710, Labor-impex). Samples were measured by the MACSQuant Analyzer 16 (Miltenyi Biotec) and analyzed by FlowJo® software (BD Biosciences). The gating strategy is outlined in Supplementary Fig. 1C.

## Statistics and reproducibility

Statistical tests, significant *p* values and number of replicates are indicated in the figure legends and are briefly described here. Immuno-peptidomics experiments were performed with four biological replicates. Mouse vaccination-challenge experiments were performed with 5 animals/group, while mouse vaccination experiments to monitor T-cell responses were performed with ten animals/group. Nonparametric Mann–Whitney, paired Student *t*, Shapiro-Wilk and Wilcoxon matched-pairs signed rank tests were performed using GraphPad Prism 9.3. *P* value thresholds used for the statistical tests corresponds to $*p < 0.05$, $**p < 0.01$, $***p < 0.001$ and $****p < 0.0001$. The values of single data points and the exact *p* values are indicated in the source data.

## Reporting summary

Further information on research design is available in the Nature Research Reporting Summary linked to this article.

## Data availability

The mass spectrometry proteomics data have been deposited to the ProteomeXchange Consortium via the PRIDE[131] partner repository with the dataset identifier PXD031451. The UniProt SwissProt and TrEMBL databases were accessed via https://www.uniprot.org/, while the IEDB database and associated analysis resource tools NetMHCpan v4.1 and MHC-NP were accessed via https://www.iedb.org/, http://tools.iedb.org/mhci/, and http://tools.iedb.org/mhcnp/, respectively. Data supporting the findings of this manuscript are available within the article, the Supplementary Data, the Supplementary Information and the Source Data files. Source data are provided with this paper.

## Code availability

The code for the calculation of correlation coefficients between *Listeria*-derived and synthetic peptides including an example peptide can be found on GitHub (https://github.com/RalfG/2022-listeria-spectrum-similarity) and Zenodo (https://doi.org/10.5281/zenodo.5948475). A runnable version of the script can be found online at Binder (https://mybinder.org/v2/gh/RalfG/2022-listeria-spectrum-similarity/HEAD?labpath=2022-listeria-spectrum-similarity.ipynb).

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

## Acknowledgements

We are grateful to Prof. Pascale Cossart for sharing the *Listeria monocytogenes* EGD strain. We thank Jarne Pauwels and Jonas Erzeel for technical assistance and the VIB Proteomics Core for LC-MS/MS analysis. R.L.M. acknowledges support from the Flemish Institute for Biotechnology (VIB) and a faculty interim fellowship from the UGent Faculty of Medicine and Health Sciences. R.V. is funded by a UGent BOF postdoc grant (BOF20/PDO/041) and A.G. is supported by a PhD fellowship from the Higher Education Commission (HEC) Pakistan. F.I. acknowledges support from Odysseus grant G0F8616N from the Research Foundation Flanders (FWO) and D.E. is funded by an FWO PhD fellowship for fundamental research. R.G. receives FWO funding (1S50918N) and F.I., L.M., B.V. and I.L. acknowledge support from Ghent University Concerted Research Action grant BOF21/GOA/033.

## Author contributions

R.M. performed the in vitro *Listeria* infection and immunopeptidomics experiments, compiled all figures and wrote and edited the paper. R.V. and I.A. performed mRNA transcription and production of mRNA Galsomes, assisted C.A. with vaccination and *Listeria* challenge assays, performed the vaccination and T-cell response measurements and wrote and edited the paper. K.B. cloned and prepared the plasmids for mRNA transcription. C.A. performed the vaccination and *Listeria* challenge assays, assisted R.V. and I.A. with the vaccination and T-cell response measurements and wrote and edited the paper. D.E. assisted R.V., I.A. and C.A. with vaccination, *Listeria* challenge and T-cell response experiments. A.G. performed in vitro *Listeria* infection and western blotting experiments to test MHC Class II expression, while I.A. performed read-out by flow cytometry. F.T. supervised A.G., generated figures and edited the paper. H.D. produced synthetic peptides for immunopeptide validation. T.M.M. assisted R.M. with immunopeptidomics data analysis and R.G. assisted with spectral validation of the synthetic immunopeptides. C.B. assisted R.M. with the conservation analysis of *Listeria* antigens and immunopeptides and edited the paper. L.M. supervised R.G. and edited the paper. S.D.S. supervised R.V. and I.A. and edited the paper. B.V. provided the W6/32 antibody for the immunopeptidomics experiments, assisted in data interpretation and edited the paper. I.L. supervised R.V. and I.A., assisted in data interpretation of the vaccination challenge and T-cell response measurements and wrote and edited the paper. F.I. conceptualized and initiated the project, supervised R.M, C.A., A.G, D.E., K.B., and F.T., interpreted the immunopeptidomics data, managed data compilation and data interpretation, generated figures, and wrote and edited the paper.

## Competing interests

The authors declare the following competing interests: R.L.M. and F.I. are inventors and R.V., C.A., I.A., S.D.S. and I.L. are contributors to patent application no. EP22170845.6, Vaccine Compositions against *Listeria* Infection. A.G., D.E., K.B., F.T., T.M.T., H.D., R.G., L.M., C.B. and B.V. declare no competing interests.

## Additional information

[1]VIB-UGent Center for Medical Biotechnology, VIB, Ghent, Belgium. [2]Department of Biomolecular Medicine, Ghent University, Ghent, Belgium. [3]VIB Proteomics Core, VIB, Ghent, Belgium. [4]Ghent Research Group on Nanomedicines, Ghent University, Ghent, Belgium. [5]Cancer Research Institute Ghent (CRIG), Ghent, Belgium. [6]Center for Medical Genetics, Ghent University Hospital, Ghent, Belgium. [7]Université Côte d'Azur, CNRS, IPMC, Sophia-Antipolis, France. [8]Department of Diagnostic Sciences, Ghent University, 9000 Ghent, Belgium. [9]Present address: Research Institute of Molecular Pathology (IMP), Vienna BioCenter, Vienna, Austria. ✉e-mail: ine.lentacker@ugent.be; francis.impens@vib-ugent.be

