## [Peer Review File · Nature Communications]

Immunopeptidomics-based design of mRNA vaccine formulations against *Listeria monocytogenes*Editorial Note: This manuscript has been previously reviewed at another journal that is not operating a transparent peer review scheme. This document only contains reviewer comments and rebuttal letters for versions considered at Nature Communications.

Reviewers' Comments:

Reviewer #1:

Remarks to the Author:

The authors have addressed all my comments with care, detail and discussion, and I have no further comments. As to my original assessment, this is a technically excellently performed study demonstrating the suitability for immunopeptidomics data to inform rational bacterial mRNA vaccine design in the context of *Listeria* infection, and it is suitable for publication in Nature Communication.

I noticed the occasional use of "epitope" before T cell reactivity of the peptide antigen was established which had escaped my previous evaluation, so this should be reviewed before publication.

Reviewer #3:

Remarks to the Author:

All of my previous concerns have been satisfactorily addressed!

Note: changes in the main text related to the reviewer comments and the items in the Author Checklist are highlighted in **green**.

Reviewer #1

The authors have addressed all my comments with care, detail and discussion, and I have no further comments. As to my original assessment, this is a technically excellently performed study demonstrating the suitability for immunopeptidomics data to inform rational bacterial mRNA vaccine design in the context of *Listeria* infection, and it is suitable for publication in *Nature Communications*.

I noticed the occasional use of "epitope" before T cell reactivity of the peptide antigen was established which had escaped my previous evaluation, so this should be reviewed before publication.

Answer: We thank the reviewer for his positive feedback and general support of our study. We have now replaced the term "epitope" with "immunopeptide" or "peptide" throughout the manuscript, except when the term was in a more broader context in the introduction or the discussion, or in the results after T-cell reactivity was established.

Reviewer #3

All of my previous concerns have been satisfactorily addressed!

Answer: We thank the reviewer for his positive feedback.